# Fidelity-Enriched Contrastive Search: Reconciling the Faithfulness-Diversity Trade-Off in Text Generation

**Wei-Lin Chen**[12*] **Cheng-Kuang Wu**[1] **Hsin-Hsi Chen**[1] **Chung-Chi Chen**[2]

[1]National Taiwan University, Taiwan
[2]Artificial Intelligence Research Center, AIST, Japan

wlchen@nlg.csie.ntu.edu.tw
hhchen@ntu.edu.tw
c.c.chen@acm.org

## Abstract

In this paper, we address the hallucination problem commonly found in natural language generation tasks. Language models often generate fluent and convincing content but lack consistency with the provided source, resulting in potential inaccuracies. We propose a new decoding method called Fidelity-Enriched Contrastive Search (FECS), which augments the Contrastive Search framework with context-aware regularization terms. FECS promotes tokens that are semantically similar to the provided source while penalizing repetitiveness in the generated text. We demonstrate its effectiveness across two tasks prone to hallucination: abstractive summarization and dialogue generation. Results show that FECS consistently enhances faithfulness across various language model sizes while maintaining output diversity comparable to well-performing decoding algorithms.[1]

## 1 Introduction

Language models (LMs) have achieved remarkable success in generating human-like text, fostering advancements across numerous Natural Language Processing (NLP) applications. Despite the fluent and seemingly convincing outputs produced by LMs, these models can occasionally generate content that is factually inconsistent with the provided source (Koehn and Knowles, 2017; Rohrbach et al., 2018; Raunak et al., 2021), an issue known as the *hallucination problem* (Maynez et al., 2020; Ji et al., 2023). Methods to mitigate hallucination have been explored from various facets, including data perspectives (Wang, 2019; Filippova, 2020; Shuster et al., 2021), model architectures (Cao et al., 2018; Aralikatte et al., 2021; Xiao and Wang, 2021), and training strategies (Huang et al., 2020; Chen et al., 2021; Li et al., 2021). In this work, we

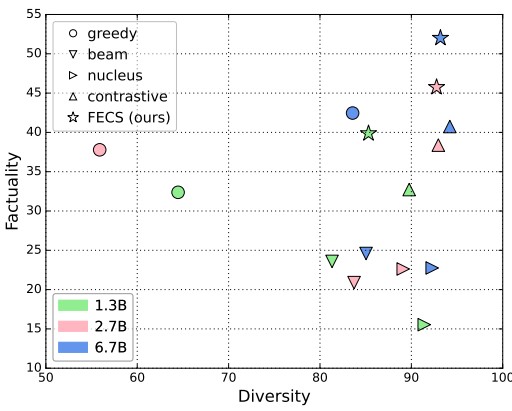

Figure 1: Results on CNN-DailyMail show our proposed FECS mitigates hallucination (i.e., improves factuality) while maintaining diversity of the generated summarization.

turn to a less investigated lens—decoding—to improve faithfulness,[2] and introduces a novel decoding method named Fidelity-Enriched Contrastive Search (FECS).

Decoding algorithms can be categorized into deterministic and stochastic groups. Deterministic methods such as beam search and greedy decoding aim to generate the most probable text continuations. While these methods might appear to be less unfaithful, they are often degenerated. That is, the outputs are uninformative, monotonous, or repetitive (Li et al., 2016; Holtzman et al., 2019; Welleck et al., 2019). Conversely, stochastic methods such as top-$k$ (Fan et al., 2018) and nucleus sampling (Holtzman et al., 2019) inject randomness into the generation process, thereby promoting the diversity. Yet, these sampling-based approaches often come at the cost of coherency and semantic consistency (Basu et al., 2020; Su et al., 2022; Su and Collier, 2023), where increasing the output diversity positively correlates with hallucinating (Dziri

---

*Work done during an internship at AIST.

[1]https://github.com/ntunlplab/FECS

[2]We follow (Ji et al., 2023) and refer to faithfulness as an antonym to hallucination, i.e., maximizing faithfulness equals minimizing hallucination.

et al., 2021). To reconcile this faithfulness-diversity trade-off, we proposed FECS—a simple yet effective decoding strategy which extends the Contrastive Search framework (Su et al., 2022) and introduces context-aware regularization terms to enhance faithfulness and penalize degeneration. Specifically, a candidate token which exhibits (1) a great semantic similarity with tokens from the provided source and (2) a low semantic similarity with previously generated tokens is rewarded with a higher score to promote its selection. Importantly, FECS can be readily applied to existing LMs off-the-shelf, without requiring further training.

We evaluate FECS on two tasks particularly prone to text hallucination: abstractive summarization and dialogue generation (Ji et al., 2023). Experimental results show that FECS consistently improves faithfulness across various LM sizes while preserving a level of diversity comparable to predominant decoding algorithms.

## 2 Methodology

In this section, we present preliminary information on Contrastive Search (Su et al., 2022) before detailing our proposed FECS.

### 2.1 Preliminary

To address shortcomings in existing decoding methods, Su et al. (2022) propose *Contrastive Search*, a new decoding approach capable of generating diverse content without compromising coherency. At time step $t$, given an input $x_{0:c+t}$, where $x_{0:c}$ signifies the prefix context and $x_{c:c+t}$ represents the previously generated tokens, Contrastive Search generates the next token $x_{c+t}$ via the following formula:

$$x_{c+t} = \arg\max_{v \in V^{(k)}} \Big\{ (1 - \alpha) \times \underbrace{p_\theta(v|x_{0:c+t})}_{\text{model confidence}} \\ - \alpha \times \underbrace{\max_{c \le j \le c+t-1} \big\{ sim(h_v, h_{x_j}) \big\}}_{\text{degeneration penalty}} \Big\}$$

Here, $V^k$ denotes a set of $k$ candidate tokens with the top-$k$ probability from the model's prediction distribution $p_\theta(\cdot|x_{0:c+t})$. The *model confidence* term represents the probability of the candidate token $v$, while the *degeneration penalty* term signifies the maximum value of the cosine similarity $sim(\cdot, \cdot)$ between candidate token $v$ and all previously generated tokens $\{x_c, ..., x_{c+t-1}\}$.

Specifically, $sim(\cdot, \cdot)$ employs the token representation $h_{x_i}$ and $h_v$ from the model's last hidden state, calculated by appending $v$ to $x_{0:c+t}$ as model input. $\alpha$ serves as a pre-determined, non-negative hyper-parameter; when $\alpha$ equals 0, Contrastive Search reduces to greedy decoding. Essentially, Contrastive Search preserves coherence by choosing outputs from the top-$k$ probable candidates while also curbing degeneration behaviors such as repetitions, thereby promoting diversity.

### 2.2 Fidelity-Enriched Contrastive Search

Motivated by Contrastive Search, we extend this framework by integrating a *faithfulness* term that encourages factuality and reduces hallucination. Using the notations from Section 2.1, we define FECS as follows:

Consider an input $x_{0:c+t}$ at time step $t$, where $x_{0:c}$ represents the prefix context, and $x_{c:c+t}$ is the previously generated tokens. We further decompose $x_{0:c}$ into: (1) the prompts $x_{0:s}$, and (2) the provided source $x_{s:c}$, which the output is expected to remain faithful to. FECS generates the next token $x_{c+t}$ via the following formula:

$$x_{c+t} = \arg\max_{v \in V^{(k)}} \Big\{ (1 - \alpha - \beta) \times \underbrace{p_\theta(v|x_{0:c+t})}_{\text{model confidence}} \\ - \alpha \times \underbrace{\max_{c \le i \le c+t-1} \big\{ sim(h_v, h_{x_i}) \big\}}_{\text{degeneration penalty}} \\ + \beta \times \underbrace{\max_{s \le j \le c-1} \big\{ sim(h_v, h_{x_j}) \big\}}_{\text{faithfulness reward}} \Big\}$$

The newly introduced faithfulness term rewards candidate tokens exhibiting high semantic similarity to tokens in the source content. Specifically, the faithfulness term denotes the maximum value of the cosine similarity $sim(\cdot, \cdot)$ between the candidate token $v$ and all source tokens $\{x_s, ..., x_{c-1}\}$. Here, $\beta$ is also a pre-determined, non-negative hyper-parameter.

## 3 Experimental Setup

### 3.1 Datasets, Models, and Configurations

We evaluate our method, FECS, on two tasks known for their susceptibility to hallucination issues: abstractive summarization and dialogue generation. For the abstractive summarization task, we adopt CNN-DailyMail (CNN-DM) dataset (Nallapati et al., 2016), a widely-used benchmark in

| Model Size | Method | CNN-DM | | | | | WoW | | | |
|---|---|---|---|---|---|---|---|---|---|---|
| | | R-1 | R-2 | R-L | BERTSc. | FEQA | B-4 | R-L | BERTSc. | Q2 |
| 1.3B | Greedy | 27.89 | 12.14 | 20.37 | 86.54 | 32.38 | 3.76 | 11.44 | 74.40 | 24.37 |
| | Beam | 28.10 | **14.14** | 20.35 | 84.34 | 23.59 | **7.65** | **17.33** | 76.51 | **36.10** |
| | Nucleus | 20.58 | 5.25 | 13.82 | 84.34 | 15.54 | 1.54 | 10.72 | 72.27 | 12.97 |
| | Contrastive | 30.06 | 11.74 | 20.80 | 86.70 | 32.73 | 4.50 | 15.89 | 74.57 | 25.42 |
| | FECS (ours) | **30.06** | 13.07 | **21.80** | **87.02** | **39.87** | 5.37 | 14.73 | **77.59** | 32.08 |
| 2.7B | Greedy | 28.61 | 12.15 | 20.99 | 86.81 | 37.78 | 4.14 | 13.33 | 70.71 | 26.39 |
| | Beam | 28.83 | **14.28** | 20.71 | 86.63 | 20.89 | 7.64 | 18.79 | **76.58** | 41.26 |
| | Nucleus | 24.48 | 7.14 | 16.73 | 85.62 | 22.62 | 1.46 | 11.19 | 72.19 | 12.60 |
| | Contrastive | **30.33** | 12.17 | 21.38 | 87.08 | 38.38 | 3.80 | 16.32 | 73.63 | 27.52 |
| | FECS (ours) | 28.74 | 12.56 | **21.45** | **87.49** | **45.75** | **9.32** | **22.42** | 75.27 | **45.10** |
| 6.7B / 6B | Greedy | 33.77 | 14.59 | 23.95 | 87.47 | 42.46 | 0.27 | 4.48 | 67.79 | 7.14 |
| | Beam | 29.99 | 14.77 | 21.18 | 86.70 | 24.59 | 0.15 | 4.46 | 74.86 | 9.15 |
| | Nucleus | 27.14 | 8.11 | 17.93 | 85.96 | 22.75 | 1.31 | 9.06 | 71.21 | 13.22 |
| | Contrastive | 33.45 | 13.08 | 23.07 | 87.33 | 40.75 | 0.87 | 9.89 | 72.60 | 14.13 |
| | FECS (ours) | **34.80** | **15.08** | **24.86** | **87.75** | **52.01** | **2.48** | **10.32** | **75.03** | **23.12** |

Table 1: Experimental results comparing FECS with other decoding methods across model scales.

several recent studies (Dong et al., 2020; Cao and Wang, 2021; Cao et al., 2020). The dialogue generation task employs the popular Wizard of Wikipedia (WoW) dataset (Dinan et al., 2018). The objective here is to generate responses based on given knowledge snippets, taken from Wikipedia, that are pertinent to the conversation topic.

In our experiments involving abstractive summarization, we adopt OPT (Zhang et al., 2022) with three scales: 1.3B, 2.7B, and 6.7B. For dialogue generation, we follow the *Few-Shot Bot* approach (Madotto et al., 2021), using GPT-Neo 1.3B and 2.7B (Black et al., 2021), along with GPT-J 6B (Wang and Komatsuzaki, 2021). All experiments are conducted with few-shot prompting, using two shots.[3] We compare FECS with Contrastive Search, Greedy Decoding, Beam Search, and Nucleus Sampling. For Beam Search, we set the beam size to 4; for Nucleus Sampling, $p = 0.95$; and for Contrastive Search, $(k, \alpha) = (4, 0.6)$. For FECS, we retain the same $\alpha$ value as Contrastive Search, setting $(k, \alpha, \beta) = (4, 0.3, 0.3)$ without hyper-parameter tuning.

## 3.2 Evaluation Metrics

Our evaluation process employs the following metrics:

**Standard Metrics.** For assessing the quality of summarization, we employ ROUGE (Lin, 2004). For dialogue generation, we use ROUGE-L and

BLEU-4 (Papineni et al., 2002). In addition, we also report BERTScore (Zhang et al., 2019) on both tasks for a more advanced soft metric.

**Faithfulness Metrics.** To measure factuality in summarization, we use FEQA (Durmus et al., 2020) following prior studies (Aralikatte et al., 2021; Chen et al., 2021). Higher FEQA scores indicate greater faithfulness of the summary to the source article. For evaluating dialogue, we employ $Q^2$ (Honovich et al., 2021), a question-answering (QA) based metric designed for assessing factual consistency in knowledge-grounded dialogue generation. Both FEQA and $Q^2$ exhibit strong correlations with human judgments.

**Diversity Metric.** For both summarization and dialogue tasks, we evaluate the diversity of the generated text $x$ by calculating

$$\text{diversity}(x) = \prod_{n=2}^{4} (1.0 - \frac{\text{Rep-}n(x)}{100})$$

where Rep-$n(x)$ measures the proportion of $n$-gram repetitions in $x$, and is calculated as

$$\text{Rep-}n(x) = (1 - \frac{|\text{unique-}n\text{-gram}(x)|}{|\text{total-}n\text{-gram}(x)|}) \times 100$$

A higher diversity score suggests the model outputs exhibit less degeneration (Welleck et al., 2019; Su et al., 2022).

---

[3]Detailed examples of prompts and additional configuration information can be found in Appendix A.

| | **Article** |
| --- | --- |

**Article**

West Ham are discussing a deal for Jamaican starlet DeShane Beckford after he impressed on trial. The skilful 17-year-old forward from Montego Bay United was invited to train with West Ham's academy earlier this month and has impressed coaches after spending two weeks with the club. Beckford also has offers from clubs in Belgium. [...] The Hammers will have the cheapest pricing strategy in the Barclays Premier League in a bid to fill the 54,000 capacity stadium when they make the switch for the 2016-17 season.

**Summary by Contrastive Search**

West Ham are discussing DeShane Beckford . Jamaican starlet impressed on trial at Upton Park .

**Summary by FECS**

West Ham are discussing a deal for Jamaican starlet DeShane Beckford . Beckford impressed on trial at West Ham earlier this month .

Table 2: An actual example of news summaries generated by Contrastive Search and FECS on an article from CNN-DailyMail. Text highlighted in green indicates factual information; red indicates hallucination not supported by the article.

## 4 Experimental Results

### 4.1 Faithfulness

Table 1 presents the results for abstractive summarization and dialogue generation. For abstractive summarization, FECS achieves substantial improvements on the factuality score across all scales, with 7.14%, 7.37%, and 9.55% increases for the 1.3B, 2.7B, and 6.7B models, respectively. Moreover, FECS records strong results in the ROUGE score and outperforms all other methods at the 6.7B scale. For dialogue generation, on the 1.3B scale, all stochastic algorithms, including FECS, fall short of Beam Search in most metrics. However, FECS surpasses other stochastic algorithms in terms of BLEU-4 and $Q^2$. Upon scaling up to 2.7B and 6B, FECS outperforms all methods substantially in terms of BLEU-4, ROUGE-L, and $Q^2$. Notably, the 6B model performs worse than its smaller counterparts, consistent with previous findings (Madotto et al., 2021).

Compared to Contrastive Search, FECS exhibits a superior ability to focus on entities within the source material, emphasizing factual information more comprehensively. As evident in Figure 2, FECS provides more complete information—comparing *"Jamaican starlet DeShane Beckford"* versus *"DeShane Beckford"*—and generates output more comprehensively, evidenced by Contrastive Search's failure to produce the time phrase *"earlier this month"*. Furthermore, when factual en-

| Dataset | Model Size | Faithfulness | Diversity |
| --- | --- | --- | --- |
| CNN-DM | 1.3B | +21.83% | -5.00% |
| | 2.7B | +19.20% | -0.20% |
| | 6.7B | +27.63% | -1.10% |
| WoW | 1.3B | +26.20% | -35.00% |
| | 2.7B | +63.88% | -11.20% |
| | 6B | +63.62% | -3.30% |

Table 3: Relative improvements in faithfulness and reduction of diversity of FECS over Contrastive Search.

tities are already present in the previous output, the degeneration penalty can inadvertently increase hallucinations. For instance, the term *"Upton Park"* produced by Contrastive Search lacks support from the source, whereas the correct output should be the previously generated *"West Ham"*. In this case, FECS accurately reproduces *"West Ham"*. Building on the framework of Contrastive Search, FECS not only inherits its properties of coherency and diversity (avoidance of degeneration) but also fosters the utilization of tokens that faithfully represent the provided source content.

### 4.2 Diversity

As we discussed in Section 1, model outputs must balance faithfulness and diversity. To better understand the impact of our proposed faithfulness reward on these two facets in the context of the original Contrastive Search, we calculated the improvements in faithfulness and the reductions in diversity based on the results from both the proposed FECS and the Contrastive Search.[4] Table 3 presents these evaluations. With the CNN-DailyMail dataset, FECS notably enhances faithfulness while marginally affecting diversity. Especially when the model size exceeds 2.7B, the decrease in diversity ranges only from 0.2% to 1.1%. These findings suggest that FECS successfully negotiates the faithfulness-diversity trade-off in abstractive summarization. Contrastingly, in the Wizard of Wikipedia dataset, FECS shows greater improvements in faithfulness and lesser reductions in diversity as the model size increases. Specifically, when the model size reaches 6.7B, FECS demonstrates a 63.62% improvement in faithfulness and experiences a mere 3.3% decrease in diversity. This implies that FECS performs more effectively when larger LMs are employed in dialogue generation tasks.

---

[4]The raw results and full evaluation for other decoding methods are provided in Table 6.

| Method | CNN-DM | | | WoW | | |
|---|---|---|---|---|---|---|
| | 1.3B | 2.7B | 6.7B | 1.3B | 2.7B | 6B |
| Greedy | 1.32 | 2.66 | 2.42 | 1.79 | 2.58 | 3.84 |
| Beam | 3.32 | 5.73 | 5.15 | 2.41 | 3.41 | 4.76 |
| Nucleus | 1.31 | 2.52 | 2.34 | 1.78 | 2.69 | 3.79 |
| Contrastive | 3.55 | 6.47 | 6.53 | 2.84 | 4.34 | 5.27 |
| FECS (ours) | 4.20 | 7.47 | 8.16 | 2.91 | 4.29 | 5.28 |

Table 4: The averaged decoding speed (sec) per instance using different decoding methods across model scales. As observed, FECS is comparable to Contrastive Search.

| Metric | Contrastive Search | | | | FECS $(\alpha, \beta)$ |
|---|---|---|---|---|---|
| | $\alpha$ | | | | |
| | 0.6 | 0.4 | 0.2 | 0.0 | (0.3, 0.3) |
| R-1 | 33.45 | 34.14 | 33.92 | 33.77 | **34.80** |
| R-2 | 13.08 | 14.17 | 14.43 | 14.59 | **15.08** |
| R-L | 23.07 | 23.91 | 23.97 | 23.95 | **24.86** |
| Diversity | **94.21** | 90.13 | 88.07 | 83.57 | 93.18 |
| FEQA | 40.75 | 41.12 | 42.37 | 42.46 | **52.01** |

Table 5: Comparison of FECS and Contrastive Search with different values of $\alpha$.

## 4.3 Analysis

**Latency.** To assess the decoding latency of our proposed FECS objective, we report the average decoding time (sec) per instance in Table 4. The results are averaged across 100 randomly selected instances. As observed in both the dialogue generation and abstractive summarization tasks, FECS and Contrastive Search perform comparably and slightly slower than beam search. Greedy and nucleus are the fastest.

**The role of $\alpha$.** To establish a more comprehensive baseline, we evaluate FECS against Contrastive Search with different values of $\alpha$ on the 6.7B model. Intuitively, a smaller $\alpha$ value (i.e., a lower degree of diversity) might contribute to a more factual performance. However, as shown in Table 5 lowering $\alpha$ only improves faithfulness marginally and with essentially the same rouge scores. On the contrary, FECS retains a high level of diversity and achieves superior performance on both FEQA and standard metrics, indicating the effectiveness of our newly introduced $\beta$ term.

## 5 Human Evaluation

In addition to the automatic evaluation, we also perform human evaluation to assess the faithfulness of our proposed FECS on the abstractive summarization task. We compare FECS against Contrastive Search, and ask annotators to vote which response is considered more faithful to the provided source (i.e., the text to be summarized). Specifically, we randomly sample 20 instance for each of the three model sizes, with a total of 60 instances for the evaluation. More details including the full evaluation protocol are provided in Appendix A.2. We present the results in Figure 2. As observed, FECS shows superior results, recording more than 60% of the votes, and outperforms Contrastive Search with more than twice the votes. The results support

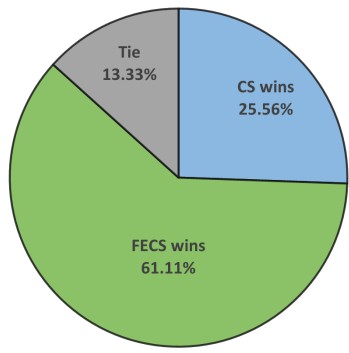

Figure 2: Human evaluation results comparing the faithfulness of FECS against Contrastive Search(CS) on the abstractive summarization task. FECS outperforms Contrastive Search, receiving more than twice the votes.

the outcome of automatic evaluation, suggesting our proposed FECS is able to generated contents which are more faithful to the provided source.

## 6 Conclusion

This paper introduces a novel decoding approach, Fidelity-Enriched Contrastive Search (FECS), designed to enhance faithfulness in text generation. Our experimental results on abstractive summarization and dialogue generation demonstrated the efficacy of FECS. It consistently improved faithfulness across various LM scales while preserving a level of diversity that is comparable to other leading decoding algorithms. Particularly when using larger LMs, it notably enhances faithfulness with only a minor impact on diversity. This indicates that FECS performs effectively when larger LMs are employed in dialogue generation tasks. In the future, we plan to explore how FECS performs with different kinds of source content, including erroneous or ambiguous inputs.

## Limitations

Firstly, while FECS presents an improvement in faithfulness and diversity trade-off, its performance

could be influenced by the quality of the source content. The assumption that source content is always correct and complete may not hold true in all scenarios, particularly in cases where the input data is ambiguous, incomplete, or erroneous. Secondly, the faithfulness assessment is primarily quantitative, based on FEQA and $Q^2$ established metrics. Although these metrics provide an essential standard for comparing models, they may not capture all nuanced aspects of faithfulness, such as the preservation of subtle implications or subjective information.

## Acknowledgments

We thank the reviewers for their insightful comments. This research was supported by JSPS KAKENHI Grant Number 23K16956 and a project JPNP20006, commissioned by the New Energy and Industrial Technology Development Organization (NEDO). This work was also partially supported by National Science and Technology Council, Taiwan, under grants MOST 110-2221-E-002-128-MY3, 110-2634-F-002-050-, and NSTC 111-2634-F-002-023-, and Ministry of Education (MOE) in Taiwan, under grants NTU-112L900901.

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

# A  Implementation Detail

## A.1  Example Prompts

Figure 3 and 4 demonstrate example prompts used in our experiments.

Figure 3: An example prompt of the CNN-DailyMail dataset for the abstractive summarzation task.

## A.2  Details of Human Evaluation

The full human evaluation protocol is presented in Figure 5. We invite three graduate-level students proficient in English for the evaluation for the annotations. As our task does not require specific domain expertise, the payment is determined by the

Figure 4: An example prompt of the Wizard of Wikipedia dataset for the dialogue generation task.

minimum wage. We also compute inter-annotator agreements by Randolph's $\kappa$, and records a moderate $\kappa = 0.57$.

| Model Size | Method | CNN-DM | | | | WoW | | | |
|---|---|---|---|---|---|---|---|---|---|
| | | Rep-2 | Rep-3 | Rep-4 | Diversity | Rep-2 | Rep-3 | Rep-4 | Diversity |
| 1.3B | Greedy | 16.22 | 12.80 | 11.75 | 64.47 | 55.33 | 54.89 | 55.21 | 9.03 |
| | Beam | 9.82 | 5.65 | 4.43 | 81.32 | 41.22 | 41.28 | 41.98 | 20.03 |
| | Nucleus | **5.33** | **2.06** | **1.41** | **91.41** | **3.31** | **1.17** | **0.63** | **94.96** |
| | Contrastive | 5.68 | 2.82 | 2.07 | 89.76 | 6.13 | 4.13 | 3.52 | 86.83 |
| | FECS (ours) | 7.60 | 4.37 | 3.45 | 85.31 | 17.91 | 17.04 | 17.08 | 56.47 |
| 2.7B | Greedy | 19.65 | 16.98 | 16.22 | 55.89 | 36.57 | 34.46 | 33.52 | 27.64 |
| | Beam | 8.72 | 4.76 | 3.69 | 83.73 | 30.35 | 29.22 | 29.05 | 34.98 |
| | Nucleus | 5.79 | 3.07 | 2.42 | 89.11 | **2.74** | **0.92** | **0.49** | **95.89** |
| | Contrastive | **4.13** | **1.82** | 1.25 | **92.95** | 3.22 | 2.03 | 1.67 | 93.23 |
| | FECS (ours) | 4.40 | 1.84 | **1.15** | 92.76 | 7.10 | 5.78 | 5.40 | 82.80 |
| 6.7B / 6B | Greedy | 8.11 | 5.09 | 4.18 | 83.57 | 45.07 | 44.22 | 44.41 | 17.03 |
| | Beam | 8.15 | 4.29 | 3.26 | 85.04 | 15.53 | 14.75 | 14.80 | 61.35 |
| | Nucleus | 4.47 | 2.03 | 1.40 | 92.28 | 2.52 | 0.83 | 0.44 | 96.25 |
| | Contrastive | **3.45** | **1.46** | **0.98** | **94.21** | **0.71** | **0.18** | **0.06** | **99.05** |
| | FECS (ours) | 4.05 | 1.76 | 1.15 | 93.18 | 2.63 | 1.07 | 0.55 | 95.80 |

Table 6: The evaluation results of repetition and diversity on FECS and other decoding methods across model scales.

Task: Abstractive Summarization

Given two summaries (*Summary_A* and *Summary_B*), you should determine which one is more faithful to the provided *Source*, and fill in "A" or "B" in the *Faithful* column.

Degree of faithfulness (from most faithful to least faithful)
1. All information presented in the summary **can be supported by the source**.
   - If there is a tie, choose the one with more correct information or is more comprehensive/complete.

2. The summary contains information which **can not be supported by the source**.
   - If there is a tie, choose the one with less information that can not be supported by the source.

3. The summary contains information which **contradicts the source**.
   - If there is a tie, choose the one with less information that contradicts the source.
4. If the two summaries are not rankable (e.g., they are exactly the same), please fill in "T" in the *Faithful* column.

Figure 5: The human evaluation protocol for the abstractive summarization task.