# OpenReview forum: "Fidelity-Enriched Contrastive Search: Reconciling the Faithfulness-Diversity Trade-Off in Text Generation"
_EMNLP/2023/Conference — EMNLP 2023 Main_

### Official Review · Reviewer_cpWD · 2023-07-24

**Typos Grammar Style And Presentation Improvements:** It is better to include a more detail…
**Soundness:** 3

**Excitement:**

3: Ambivalent: It has merits (e.g., it reports state-of-the-art results, the idea is nice), but there are key weaknesses (e.g., it describes incremental work), and it can significantly benefit from another round of revision. However, I won't object to accepting it if my co-reviewers champion it.

**Missing References:**

N/A

**Paper Topic And Main Contributions:**

The paper focuses on the hallucination problem in existing decoding methods, i.e., SimCTG. To address the problem, authors propose the fidelity-enriched contrastive search (FECS) to reconcile faithfulness and diversity. The core idea of FECS is to select the token having maximum similarities to the source context and minimal similarities to the already generated tokens. The method is evaluated on abstractive summarization and dialogue response generation tasks. Experimental results show that FECS achieves a good trade-off on faithfulness and diversity.

**Questions For The Authors:**

Question A. Have you compared the decoding latency between FECS and SimCTG?

Question B. Have you attempted more prompts? Does the conclusion still stand with different prompts?


**Reasons To Accept:**

1. Good motivation for exploring the hallucination problem in the existing decoding methods.

2. Good presentation.

3. A straightforward and effective method is proposed.

**Reasons To Reject:**

1. Incorrect equation used in Section 2.1. SimCTG is to minimize the similarity between the candidate token with **ALL** previous tokens, rather than the previously generated tokens. It should be $0 \lt j \le c+t-1$ in the degeneration penalty term.

2. The selected evaluation metrics are not convincing. We have ever conducted many experiments on SimCTG. We observed that SimCTG usually has worse performance on the traditional text-matching-based metrics, like Rouge and BLEU. So, I'd like to see the comparison results of your model against SimCTG on the more advanced evaluation methods, like MAUVE, BertScore, or *human evaluation*.

3. Hyper-parameter search is missing. Firstly, the workload of hyper-parameter search is very heavy in this work since you additionally introduced a new parameter $\beta$. But, at least, I think authors should explore how $\alpha$ and $\beta$ respectively affect the tradeoff between faithfulness and diversity with the value of $k$ fixed.

4. Apart from the direct inference in the manner of *few-shot bot* based on LLMs, I am also glad to see the improvements in the decoding stage after fine-tuning the models, like the way in SimCTG. But consider that this is a short paper. So, the current evaluation only on LLMs is also enough.


**Reproducibility:**

4: Could mostly reproduce the results, but there may be some variation because of sample variance or minor variations in their interpretation of the protocol or method.

**Reviewer Confidence:**

5: Positive that my evaluation is correct. I read the paper very carefully and I am very familiar with related work.

---

> ### Author Rebuttal · Authors · 2023-08-28
>
> Thank you for the constructive feedback! To alleviate your concerns raised in “Reasons To Reject”, we first provide the necessary background information:
>
> - *SimCTG* is a fine-tuning framework for calibrating the representation space of ***anisotropic*** autoregressive LMs—LMs which suffer the degeneration problem since their token representations reside in a narrow subset of the entire space. After fine-tuning with *SimCTG*, the LMs become ***isotropic***, and the Contrastive Search decoding strategy can be applied to LMs to obtain degeneration-free results.
>
> - *SimCTG* and Contrastive Search is first proposed in [1]. Later on, the same authors published another paper [2], in which they reinvestigated the problem *“Are autoregressive LMs really anisotropic?”* and find that except *GPT-2* small/medium, most autoregressive are in fact isotropic, thus, Contrastive Search can be used on off-the-shelf autoregressive LMs without *SimCTG* fine-tuning.
>
> - Reference
>    - [1] Su, Y., Lan, T., Wang, Y., Yogatama, D., Kong, L., & Collier, N. (2022). A contrastive framework for neural text generation. Advances in Neural Information Processing Systems, 35, 21548-21561.
>    - [2] Su, Y., & Collier, N. (2022). Contrastive Search Is What You Need For Neural Text Generation. Transactions on Machine Learning Research
>
> > Incorrect equation used in Section 2.1. *SimCTG* is to minimize the similarity between the candidate token with ALL previous tokens, rather than the previously generated tokens. It should be $ 0 < j <= c + t - 1 $ in the degeneration penalty term.
> - Though the definition presented in [1][2] shows that Contrastive Search considers all previous tokens, in the few-shot prompting (in-context learning) setting, only the generated tokens are considered in Contrastive Search’s implementation --- see Line 48 ```block_context_degeneration_penalty = True``` in their official GitHub repo: github.com/yxuansu/Contrastive_Search_Is_What_You_Need/blob/542d6ecd7b99f3bc04a514d89b99bbfcc707bbb4/summarization/inference.py#L48C13-L48C47.
>
> > The selected evaluation metrics are not convincing. We have ever conducted many experiments on SimCTG. We observed that SimCTG usually has worse performance on the traditional text-matching-based metrics, like Rouge and BLEU. So, I'd like to see the comparison results of your model against SimCTG on the more advanced evaluation methods, like MAUVE, BertScore, or human evaluation.
> - Thanks for the suggestion and the provided insights, we will investigate suitable, advanced NLG metrics, including *MAUVE* and *BertScore*, and report results in our final version for a more comprehensive evaluation.
>
> > Hyper-parameter search is missing. Firstly, the workload of hyper-parameter search is very heavy in this work since you additionally introduced a new parameter beta. But, at least, I think authors should explore how alpha and beta respectively affect the tradeoff between faithfulness and diversity with the value of fixed.
> - Indeed, introducing beta substantially increases the hyper-parameter search workload, and we agree a further analysis of the effect on faithfulness and diversity under different alpha-beta pairs is beneficial, and will be incorporated in the camera-ready version.
>
> > Apart from the direct inference in the manner of few-shot bot based on LLMs, I am also glad to see the improvements in the decoding stage after fine-tuning the models, like the way in SimCTG. But consider that this is a short paper. So, the current evaluation only on LLMs is also enough.
> - As detailed in the above necessary background information, Su et al. suggest most autoregressive LMs are in fact isotropic, thus, Contrastive Search is directly applicable to out-of-the-box autoregressive LMs without *SimCTG* fine-tuning.
>
> Regarding the “Questions For The Authors”:
> > Have you compared the decoding latency between FECS and SimCTG?
> - Following, we report the average decoding time (sec) per instance for different decoding algorithms. The results are averaged across 100 randomly selected instances. As observed in both the dialogue generation and abstractive summarization tasks, FECS and Contrastive Search CS perform comparably and slightly slower than beam search. Greedy and nucleus are the fastest.
>
> - Dialogue Generation
>
>    |      | Greedy | Beam | Nucleus |  CS  | FECS |
>    | ---- |:------:|:----:|:-------:|:----:|:----:|
>    | 1.3B |  1.79  | 2.41 |  1.78   | 2.84 | 2.91 |
>    | 2.7B |  2.58  | 3.41 |  2.69   | 4.34 | 4.29 |
>    | 6.B  |  3.84  | 4.76 |  3.79   | 5.27 | 5.28 |
>
> - Abstractive Summarization
>    |      | Greedy | Beam | Nucleus |  CS  | FECS |
>    | ---- |:------:|:----:|:-------:|:----:|:----:|
>    | 1.3B |  1.32  | 3.32 |  1.31   | 3.55 | 4.20 |
>    | 2.7B |  2.66  | 5.73 |  2.52   | 6.47 | 6.53 |
>    | 6.7B  |  2.42  | 5.15 |  2.34   | 7.47 | 8.16 |
>
> > Have you attempted more prompts? Does the conclusion still stand with different prompts?
> - We adopt the same prompts as the prior works without prompt search/engineering. We acknowledge that it will be beneficial to experiment with a collection of prompt variations to further strengthen the effectiveness of FECS, and it is left as our future work.
>
> Regarding the "Typos Grammar Style And Presentation Improvements":
>
> > It is better to include a more detailed explanation for Figure 1.
> - Thanks for pointing this out! Figure 1 presents a visualization of the experimental results of the abstractive summarization task (i.e., CNN-DailyMail). We will add a detailed caption in the final version.

---

### Official Review · Reviewer_evYM · 2023-08-02

**Soundness:** 4

**Excitement:**

4: Strong: This paper deepens the understanding of some phenomenon or lowers the barriers to an existing research direction.

**Paper Topic And Main Contributions:**

- Decoding strategy for enhancing faithfulness without loss of the diversity
- Fidelity-enriched contrastive search that even shows high faithfulness than the conventional contrastive search algorithm

**Questions For The Authors:**

- Is there any specific reason that similarity is estimated with last hidden states? It seems that similarity between logits would be more intuitive approach.

- Have you tried experiments on the alpha / beta variants? It seems that by adjusting alpha value, contrastive setting would give competitive performance with FECS. Perhaps (k, α, β) = (4, 0.3, 0.3) and (k, α, β) = (4, 0.6) may not be the representative performance with respect to the faithfulness.

**Reasons To Accept:**

- Clearly stated the main objective and experimental results properly support it
- Substantially enhanced factuality by simple and effective approach


**Reasons To Reject:**

Nothing special

**Reproducibility:**

5: Could easily reproduce the results.

**Reviewer Confidence:**

4: Quite sure. I tried to check the important points carefully. It's unlikely, though conceivable, that I missed something that should affect my ratings.

---

> ### Author Rebuttal · Authors · 2023-08-28
>
> Thank you for the positive comments! Regarding the “Questions For The Authors”:
>
> > Is there any specific reason that similarity is estimated with last hidden states? It seems that similarity between logits would be more intuitive approach.
>
> - We use the hidden states to compare similarities following the original Contrastive Search implementation. A potential issue of using logit is the computation overhead, as logit size equals vocabulary size. Moreover, the dense vector representations of hidden states could be better in capturing tokens’ meaning in comparison to logits, which could be very sparse.
>
> > Have you tried experiments on the $\alpha$ /$\beta$ variants? It seems that by adjusting alpha value, the contrastive setting would give competitive performances with FECS. Perhaps $(k, \alpha, \beta)$ = $(4, 0.3, 0.3)$ and $(k, \alpha)$ = $(4, 0.6)$ may not be the representative performance with respect to the faithfulness.
>
> - Thanks for bringing this up! Following, we report results comparing FECS with Contrastive Search under different α values. We experiment on the 6.7B model for the abstractive summarization task, and $k$ = $4$ for all settings.
>
>    |    | R-1    | R-2    | R-L    | Diversity | FEQA |
>    | ---------- | --- | --- | --- | -------- | -------- |
>    | CS, α=0.6 | 33.45    | 13.08    | 23.07    | **94.21**         | 40.75         |
>    | CS, α=0.4 | 34.14    | 14.17    | 23.91    | 90.13         | 41.12         |
>    | CS, α=0.2 | 33.92    | 14.43    | 23.97    | 88.07         | 42.37         |
>    | CS, α=0.0           | 33.77    | 14.59    | 23.95    | 83.57         | 42.46         |
>    | FECS, (α,β)=0.3 | **34.80**    | **15.08**    | **24.86**    | 93.18     | **52.01**     |
>
> - As observed, lowering α notably sacrifices the diversity of Contrastive Search. Moreover, it only brings a marginal boost to the faithfulness metric (FEQA) with essentially the same ROUGE scores. On the contrary, FECS retains a high level of diversity and achieves superior performance on both the faithfulness metric and ROUGE scores.

---

### Official Review · Reviewer_2F8i · 2023-08-04

**Soundness:** 4

**Excitement:**

4: Strong: This paper deepens the understanding of some phenomenon or lowers the barriers to an existing research direction.

**Paper Topic And Main Contributions:**

This short paper proposes a new decoding objective in order to tackle the problem of generating both faithful and diverse text. It is inspired by Contrastive Search, which uses a similarity term as the decoding objective, besides the generation probability term, to penalize semantic repetition of the generated text. This paper in addition proposes another similarity term which encourages semantic alignment to the source.

The evaluation is conducted on two tasks prone to hallucination: abstractive summarization and dialogue generation. Results show that the proposed method can consistently improve faithfulness without sacrificing too much the output diversity, especially when the model size is large.

**Questions For The Authors:**

* Is there a special reason to use probability, instead of log-probability, as the model confidence in the definition of decoding objective?

* How does the new objective impact the decoding speed?

**Reasons To Accept:**

This is a well-written paper which tackles the important problem of generating faithful text without sacrificing diversity. The proposed method is straightforward and the evaluation is solid.

**Reasons To Reject:**

Some human evaluation of the faithfulness and diversity, and more examples of generated text, would make the paper more convincing.

**Reproducibility:**

4: Could mostly reproduce the results, but there may be some variation because of sample variance or minor variations in their interpretation of the protocol or method.

**Reviewer Confidence:**

4: Quite sure. I tried to check the important points carefully. It's unlikely, though conceivable, that I missed something that should affect my ratings.

---

> ### Author Rebuttal · Authors · 2023-08-28
>
> Thank you for the positive feedback! Regarding the “Reasons To Reject”:
>
> > Some human evaluation of the faithfulness and diversity, and more examples of generated text, would make the paper more convincing.
> - A human evaluation of the faithfulness and diversity of the generated results will be scheduled, and more non-cherry-picked examples will be added in our final version. Thanks for the suggestion!
>
> Regarding the “Questions For The Authors”:
>
> > Is there a special reason to use probability, instead of log-probability, as the model confidence in the definition of decoding objective?
> - We follow Contrastive Search (CS) for the use of probability. Although it was not explicitly stated in the CS paper, we hypothesize the use of probability is to better match the scale of the penalty term, $[-1, 1]$, calculated by cosine similarity. In our preliminary experiments, we also try normalizing FECS’s degeneration penalty and faithfulness reward terms from $[-1, 1]$ to $[0, 1]$, and the results are nearly identical to no normalization. Therefore, we adopt the original CS implementation for FECS.
>
> > How does the new objective impact the decoding speed?
> - Following, we report the average decoding time (sec) per instance, where the results are averaged across 100 randomly selected instances. As observed in both the dialogue generation and abstractive summarization tasks, FECS and CS perform comparably and slightly slower than beam search. Greedy and nucleus are the fastest.
>
> - Dialogue Generation
>
>    |      | Greedy | Beam | Nucleus |  CS  | FECS |
>    | ---- |:------:|:----:|:-------:|:----:|:----:|
>    | 1.3B |  1.79  | 2.41 |  1.78   | 2.84 | 2.91 |
>    | 2.7B |  2.58  | 3.41 |  2.69   | 4.34 | 4.29 |
>    | 6.B  |  3.84  | 4.76 |  3.79   | 5.27 | 5.28 |
>
> - Abstractive Summarization
>    |      | Greedy | Beam | Nucleus |  CS  | FECS |
>    | ---- |:------:|:----:|:-------:|:----:|:----:|
>    | 1.3B |  1.32  | 3.32 |  1.31   | 3.55 | 4.20 |
>    | 2.7B |  2.66  | 5.73 |  2.52   | 6.47 | 6.53 |
>    | 6.7B  |  2.42  | 5.15 |  2.34   | 7.47 | 8.16 |

---

### Meta-Review · Area_Chair_NsFb · 2023-09-18

**Recommendation:** 4

**Metareview:**

This paper introduces a new decoding method by extending the popular contrastive search with new regularization terms to improve the faithfulness-diversity tradeoff. The reviewers largely find the method to be new, simple, straightforward and well motivated. The reviewers also find the writing to be very clear. The reviewers have shared a bunch of concerns. Some of the concerns such as choice of probability measure, choice of hidden states over logits, decoding speed comparison were adequately answered by the authors. However, there are a number of concerns such as lack of human evaluation, sensitivity of the results to prompts, missing important evaluation metrics and hyperparameter search, which were unaddressed and could impact the assessment of this work. These should be addressed in the revision to maximize the impact of this work.

---

### Decision · Program_Chairs · 2023-10-07

**Decision:**

Accept-Main

**Comment:**

This paper introduces a new decoding method by extending the popular contrastive search with new regularization terms to improve the faithfulness-diversity tradeoff. The reviewers largely find the method to be new, simple, straightforward and well motivated. The reviewers also find the writing to be very clear. The reviewers have shared a bunch of concerns. Some of the concerns such as choice of probability measure, choice of hidden states over logits, decoding speed comparison were adequately answered by the authors. However, there are a number of concerns such as lack of human evaluation, sensitivity of the results to prompts, missing important evaluation metrics and hyperparameter search, which were unaddressed and could impact the assessment of this work. These should be addressed in the revision to maximize the impact of this work.